# Characteristics of Visual Saliency Caused by Character Feature for Reconstruction of Saliency Map Model

**Hironobu Takano *** , **Taira Nagashima and Kiyomi Nakamura**

Graduate School of Engineering, Toyama Prefectural University, 5180 Kurokawa, Imizu, Toyama 939-0398, Japan; nagashima@neu.pu-toyama.ac.jp (T.N.); nakamura@neu.pu-toyama.ac.jp (K.N.)
* Correspondence: takano@pu-toyama.ac.jp; Tel.: +81-766-56-7500

**Abstract:** Visual saliency maps have been developed to estimate the bottom-up visual attention of humans. A conventional saliency map represents a bottom-up visual attention using image features such as the intensity, orientation, and color. However, it is difficult to estimate the visual attention using a conventional saliency map in the case of a top-down visual attention. In this study, we investigate the visual saliency for characters by applying still images including both characters and symbols. The experimental results indicate that characters have specific visual saliency independent of the type of language.

**Keywords:** visual attention; visual saliency; character; fixation ratio

## 1. Introduction

Information technology has entered a new phase due to the drastic development of artificial intelligence. The relationships between humans and machines are also changing. Although we have provided instructions to machines unilaterally, an intelligent and autonomous machine that can detect the intentions and emotions of humans is required. Gaze information is effective in estimating the internal conditions of human beings, e.g., emotion, attention, and intention. This study focuses on the visual attention derived from the gaze information. Visual attention is divided into two categories: top-down and bottom-up attention. Bottom-up attention is defined by the attention to a visually salient region that is represented by low-level features, such as the intensity, orientation, and color. Top-down attention is task-driven and dependent on prior knowledge of each individual.

Gaze information is obtained by directly measuring gaze direction using an eye tracking device. In another method, the attention region is estimated without gaze measurement. A saliency map model used to estimate the visual attention of humans without gaze information was proposed. The saliency map is constructed by using physical features (intensity, orientation, and color), and it accurately estimates bottom-up visual attention [1]. A plethora of saliency map models that are based on not only bottom-up attention but also on top-down attention or the fusion of both types has been proposed. A graph-based saliency map (GBVS) is constructed by introducing a fully connected graph into Itti's model [2]. The proto-object saliency map model employs a biologically plausible feature [3,4]. The hybrid model of visual saliency was developed by using low, middle, and high-level image features [5]. In recent years, deep learning based models have been proposed. These models express the characteristics of bottom-up attention [6,7], hybrid bottom-up and top-down attention [8], eye fixation [9], and sequential eye movements [10,11]. However, instead of an image feature, a meaning map that utilizes meaning information in a scene has been proposed [12]. Comparing the prediction accuracy of human attention obtained from the meaning map and GBVS, the spatial distribution of attention is similar in both the methods. However, by controlling the relationship between meaning and saliency, the meaning map sufficiently expresses attention guidance compared with GBVS. In this study, the model derived using low-level features is dealt with as a saliency map.

Various applications employ saliency maps to develop an interactive system [13,14]. These applications require the use of highly accurate visual saliency maps. However, the visual saliency map does not always provide an accurate estimation result because the visual attention of human beings consists of both top-down and bottom-up attention. To overcome this problem, saliency map models, involving the use of the top-down attention effect, have been proposed [15]. A human face with high-level semantic features can be considered a salient object. Cerf et al. proposed a novel saliency map model combining face detection and bottom-up attention in the low-level visual information processing of human beings [16]. Notably, the estimation accuracy of visual attention significantly increases when using the proposed model.

In this study, we focus on the features of the characters. The conventional saliency map model proposed by Itti et al. cannot estimate visual attention for text in an advertisement. Due to the fact that characters contain high-level semantic information, they cannot always be considered as the objects that are represented by using bottom-up saliency. In this study, we considered that the attention paid to the characters exists between bottom-up and top-down attention. We attempted to develop a saliency map model for the character features, and we propose that it should include saliency characteristics. In order to reconstruct the saliency map model, we investigated whether character features have visual saliency, which induces human attention such as human faces. We conduct an experimental evaluation using scenery images including Japanese Hiragana, English alphabet letters, Thai characters, and simple symbols as visual stimuli. The subjects participating in the experiments were Japanese students. As such, the English alphabet was familiar to them, although foreign. By contrast, they found Thai characters to be unfamiliar. We herein discuss the relationship between visual attention and familiarity by using these characters.

The rest of this paper is organized as follows. Section 2 describes the related studies including their physiological findings. Section 3 presents the experimental method and results. In Section 4, we discuss visual saliency for characters from the experimental results. Finally, we describe our conclusions in Section 5.

## 2. Related Work

### 2.1. Visual Attention

Humans continually select the information from the huge amounts of information they receive in their daily life. Attention plays a crucial role in information selection. Human attention can be classified into four categories: focused attention, divided attention, anticipation-expectancy, and selective attention. In this paper, we focus on the selective attention in visual perception. The recognition of natural and familiar objects and the search for frequent stimulations have been conducted using a hardwired binding process that occurs without attention [17]. In the task of face recognition or gender identification from faces, the importance of attention is low for task implementation because the face is considered a familiar object [18,19]. Since we gain knowledge of characters through training over time, the characters are assumed to be familiar objects. Thus, we consider that character recognition is performed through a hardwired process, which does not always involve attention. In other words, attention to characters is not necessarily described as a saliency map obtained from low-level features.

### 2.2. Saliency Map Model

A saliency map is constructed by normalizing and integrating the differential images obtained by extracting physical features of an image based on intensity, orientation, and color [1]. The visual saliency map model is adapted to a dynamic image by combining the features of intensity variation and moving direction with the original image features [20]. The visual saliency map is used as an image segmentation, which indicates the classification between the foreground and background [21]. In addition, a saliency map is also adopted as text detection in natural scenes [22].

## 2.3. Visual Saliency for Characters

Visual saliency for characters has been investigated by using images, including human faces, mobile phones, and characters [23]. The experimental results have indicated that the characters are more salient in comparison with mobile phones. It was also found that faces are the most salient objects among the visual stimuli that were used in this experiment. Wang et al. investigated visual attention for characters using scenery images with alphabet letters inserted [24]. This study also indicated the visual saliency of characters. In this study, visual saliency for the English alphabet was investigated, and native English speakers were employed as subjects. Visual saliency for each character differs from each language because the effect of a particular language depends on its culture. In addition, the visual saliency for familiarity with characters was not evaluated. In the present study, the visual saliencies of Japanese characters (Hiragana), the English alphabet as a familiar foreign language writing system, and Thai characters as an unfamiliar foreign language for Japanese people were investigated by using scenery images, including simultaneously inserted characters and simple symbols.

## 3. Experiment

In order to investigate whether characters induce visual attention, experiments were conducted by randomly presenting target and non-target stimuli to the subjects. The target images included both characters (Japanese Hiragana, alphabet letters, or Thai characters) and simple symbols. An example of the characters and symbols inserted into visual stimuli is shown in Figure 1. Figure 1a–d shows Hiragana, alphabet letters, Thai characters, and simple symbols. As shown in this figure, these characters and symbols are classified into different categories because their shapes are quite different from each other. A non-target image is an original image without characters or simple symbols. The visual saliency of the characters was estimated by making a comparison between the visual fixation ratios of characters and simple symbols.

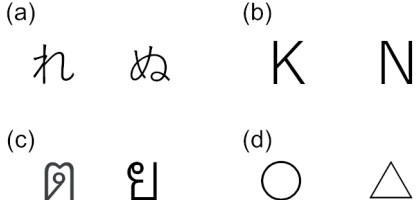

**Figure 1.** Examples of inserted characters and symbols: (**a**) Hiragana, (**b**) alphabet letters, (**c**) Thai characters, and (**d**) simple symbols.

## 3.1. Experiment Method

We describe the experiment settings, conditions including the experiment procedure, and visual stimuli presented to the subjects. We also describe how visual stimuli were prepared.

### 3.1.1. Experiment Settings

The gaze of the subjects was measured using an eye tracker (Tobii X2-30, sampling rate of 30 Hz) during the presentation of the target and non-target stimuli. A 21.5-inch PC display was used for the visual stimulus presentation, and the distance between the subject and the PC display was approximately 60 cm. The pixel resolution of the display used for the visual stimulus is 1920 × 1080. The pixel resolution of the visual stimulus is 1280 × 960. A visual stimulus was presented in the center of the display. The subject sat on a chair with a relaxed posture during the experiment. The external stimulus was removed by surrounding the subject with partition boards.

### 3.1.2. Experiment Conditions

During the experiment with Japanese Hiragana, alphabet letters, and simple symbols inserted as the visual stimuli, 10 students (eight males and two females) in their twenties participated in the experiment. During the experiment with Japanese Hiragana, Thai characters, and simple symbols inserted as visual stimuli, 16 students (11 males and 5 females) in their twenties participated. The native language of the subjects was Japanese. All the subjects could read and write in English. Although the subjects could not recognize Thai characters as being Thai, they recognized them as characters of a particular language rather than symbols. However, the individual difference in familiarity between the characters and symbols could not be confirmed from the results of the questionnaire. We did not inform the subjects of the objective of this experiment. The institutional ethics committee approved this experiment, and the subjects provided informed consent prior to participation.

Five image datasets were used as visual stimuli in the experiment. The target stimuli in Dataset 1 consist of images including two meaningless Japanese Hiragana and two simple symbols. The target images in Dataset 2 have two meaningless alphabet letters instead of Hiragana. In Dataset 3, the target stimuli include both Hiragana and alphabet letters. Dataset 4 has two meaningless Thai characters and simple symbols as the inserted target images. The target images with two meaningless Japanese Hiragana and Thai characters were included in Dataset 5. Each dataset has 50 target images and 50 non-target images. Among Datasets 1–5, the images used as visual stimuli are of the same combination.

The subjects conducted two sessions for each dataset. Thus, each subject conducted a total of six sessions with Datasets 1–3 and four sessions with Datasets 4 and 5. The sequential order of the experimental sessions was randomly selected for each subject. In one session, 25 target images and 25 non-target images were presented to the subjects. Visual stimuli were randomly presented to the subjects for a period of 2 s. The subjects were instructed to freely view an image presented in the display without a specific task. Under the free-viewing task, the attention of the subject was consistent with the fixation point [25]. In addition, except for the character features, the top-down attention induced could be suppressed as much as possible.

### 3.1.3. Visual Stimuli

Two types of visual stimuli were employed as the target and non-target images for the experiment. The target image was created by inserting the characters or symbols into the original image. The non-target image is the original image without image processing. The original image was selected from the image database of the International Affective Picture Systems (IAPS) [26]. Using Self-Assessment Manikin, the images in the IAPS were evaluated with respect to nine grades among three criteria (valence, arousal, and dominance). In this study, the evaluation values of 100 images selected as the visual stimuli were from 3 to 7 for valence and from 1 to 5 for arousal. In order to suppress individual variation in the top-down attention feature caused by emotions that are induced, we adopted these image selection criteria. The 100 images selected were randomly separated into 50 target and 50 non-target images.

Two types of objects, i.e., character and symbol, were simultaneously inserted into the target image. The characters and symbols were inserted into the same regions of an original image that was commonly stored in all five datasets. An example of a visual stimulus is shown in Figure 2. Figure 2a shows the target image with both Hiragana and simple symbols in Dataset 1. Figure 2b shows the target image with the insertion of both alphabet letters and simple symbols in Dataset 2. The low-level feature properties of the characters and symbols may give rise to differences in the characteristics of attention. Therefore, the sizes of the characters and symbols inserted into the stimulus image are sufficiently small compared with the size of the stimulus image itself. Blurring was applied around the region of insertion of the characters and symbols in order to reduce the artificiality. After inserting the characters and symbols, visual saliency was calculated by using a saliency map model. The image was used as the target stimulus when the inserted characters and symbols were not included in the top 25% of the salient regions.

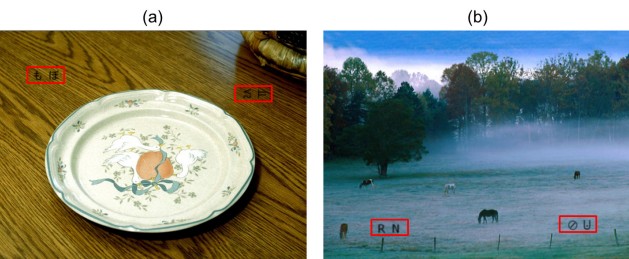

**Figure 2.** Examples of visual stimuli in (**a**) Dataset 1 and (**b**) Dataset 2. Inserted characters and symbols are visible in the red square. The red square is not included in the actual visual stimuli.

### 3.2. Analytical Method

Posner showed that a gaze fixation region is not necessarily consistent with a visual attention region [27]. However, subjects rarely turn their attention to another region that differs from the fixation point because they can freely view the stimulus image [25]. In addition, visual information cannot be perceived owing to saccadic suppression [28]. Thus, in this study, we simply measured the eye movement that occurred during anovert attention.

Visual fixation was determined when the measured gaze points remained within a certain spatial range within a certain period. The decision time of the visual fixation was fixed at 100 ms. The spatial range for a decision of the visual fixation was determined based on the variations in the gaze measurement, which was obtained during the calibration process. Thus, the spatial range for judgment as a visual fixation differed for each subject. In the current experiments, the criterion of the spatial range was determined within the range of 30 to 100 pixels.

Visual attention on the inserted characters or symbols was judged when the fixation occurred in the square region surrounding the characters or symbols, indicated by red squares in Figure 3. The region used for detecting visual attention on the characters or symbols was obtained by taking into consideration the calibration results of a gaze measurement. In Figure 3, the black square represents the minimum square region surrounding the characters or symbols. The red square regions represent the fixation decision region. The red square region was determined by using the calibration data for each subject.

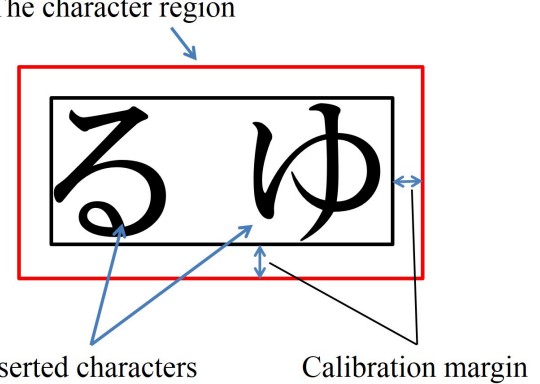

**Figure 3.** Character region including the calibration margin.

In a previous study, the degree of visual saliency was estimated by using the frequency and order of the fixations [16,23,24]. In the current study, the visual saliency of the characters or symbols was investigated based on the order of fixations and cumulative fixation, as shown in [16,23].

### 3.3. Results

Figures 4–6 depict fixation ratios as a function of fixation order using Datasets 1–3, respectively. In these figures, the horizontal axis is the order of the first visual fixation on the characters or simple symbols. The vertical axis is the ratio of the fixation on the

characters or symbols. The bar shows the visual fixation ratio for the orders of the first fixation. The solid line indicates the cumulative fixation ratio, and the values shown in the upper-right corner are the cumulative fixation ratios. In this experiment, the total number of target stimuli was 500 (50 times per subject). The numerator represents the number of times the gaze point stays around the characters or symbols. When the gaze point stays around the characters or symbols in the stimulus image at least once, the numbers of numerators are summed. Note that the repeated fixations on the characters or symbols are not counted.

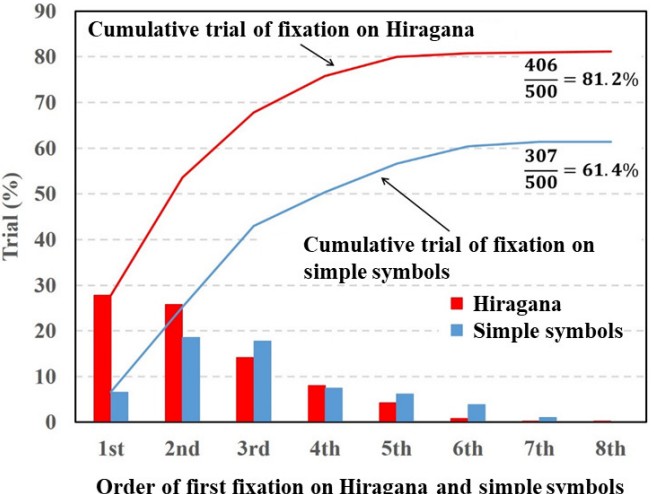

**Figure 4.** Visual fixation ratios on Hiragana and simple symbols.

From the experiment results shown in Figure 4, the fixation ratio of the Hiragana was the highest in the first order of fixation and gradually decreased with the fixation order. By contrast, the fixation ratio of the simple symbols was the highest in the second order of fixation. The total fixation ratio of the Hiragana was higher than that of the simple symbols. A paired *t*-test revealed a significant difference in fixation between Hiragana and simple symbols ($p = 0.0033$, $t(9) = 3.945$, $d = 1.09$).

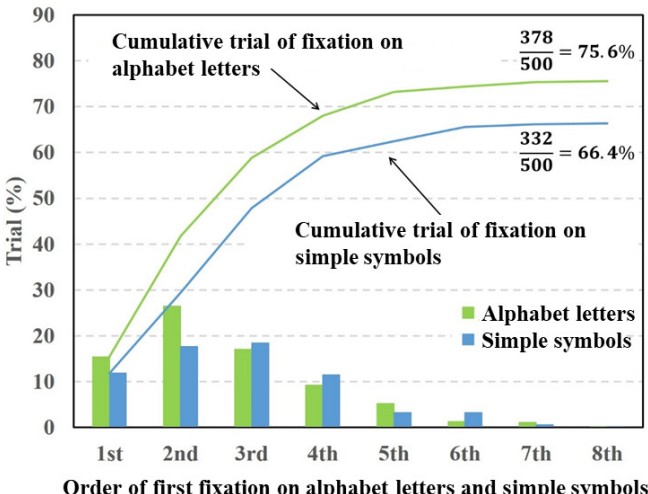

**Figure 5.** Visual fixation ratios on alphabet letters and simple symbols.

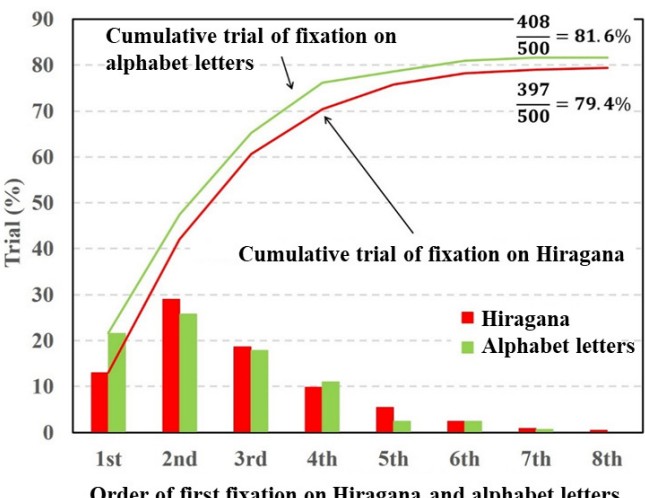

**Figure 6.** Visual fixation ratios on Hiragana and alphabet letters.

In Figure 5, the fixation ratios of the alphabet letters showed a maximum at the second order of fixation. By contrast, the fixation ratio of the simple symbols gradually increased and declined from the third order of fixation. The total fixation ratio of the alphabet letters was higher than that of the simple symbols. The paired *t*-test was conducted on the total fixation ratios between the alphabet letters and simple symbols. The result of the *t*-test showed a significant difference in fixation between the alphabet letters and simple symbols ($p = 0.0057$, $t(9) = 3.607$, $d = 0.49$).

Figure 6 shows that the fixation ratios of the alphabet letters and Hiragana had maximum values at the second order of fixation and decreased with respect to the fixation order. The total fixation ratios of the alphabet letters and Hiragana were almost same. The paired *t*-test was conducted on the total fixation ratios between the alphabet letters and Hiragana. The result of the *t*-test showed no significant difference in fixation between the alphabet letters and Hiragana ($p = 0.247$, $t(9) = -1.239$, $d = 0.12$).

Figures 7 and 8 show the fixation ratios as a function of the fixation order using Datasets 4 and 5, respectively. In Figure 7, the fixation ratio of the Thai characters decreased from the third order of fixation. By contrast, the fixation ratio of the simple symbols reached a maximum at the second order of fixation and then gradually decreased. The total fixation ratio of the Thai characters was higher than that of the simple symbols. The paired *t*-test was applied to the total fixation ratios between the Thai characters and simple symbols. The result of the *t*-test showed a significant difference in fixation between the Thai characters and simple symbols ($p = 0.00088$, $t(15) = 4.134$, $d = 0.45$).

Figure 8 shows that the fixation ratios of the Thai characters and Hiragana in the first and second orders of fixation were almost the same and decreased from the third order of fixation. The total fixation ratios of Hiragana and Thai characters was almost the same. The paired *t*-test was conducted on the total fixation ratios between the Hiragana and Thai characters. The result of the *t*-test showed no significant difference in terms of fixation between Hiragana and Thai characters ($p = 0.406$, $t(15) = 0.856$, $d = 0.12$).

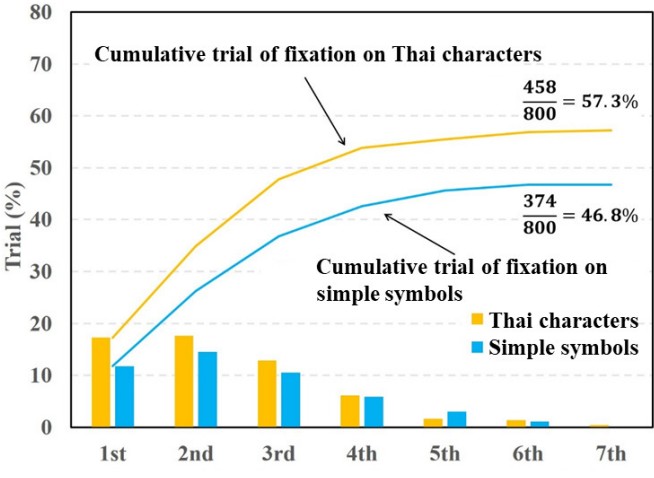

**Figure 7.** Visual fixation ratios on Thai characters and simple symbols.

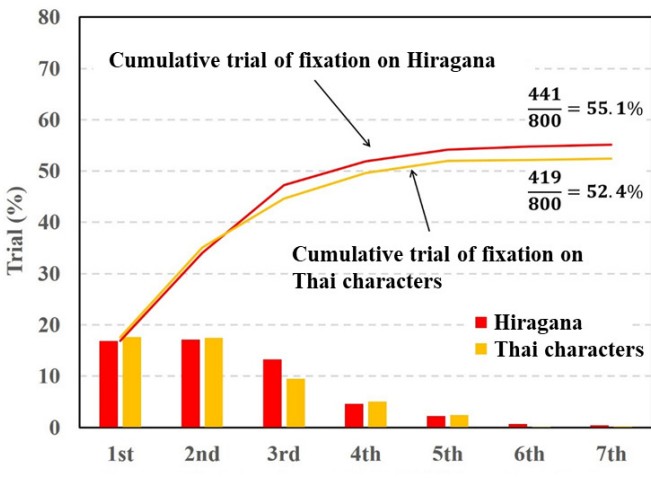

**Figure 8.** Visual fixation ratios on Hiragana and Thai characters.

## 4. Discussion

From the experimental results shown in Figures 4, 5 and 7, the fixation ratios of the characters (Hiragana, alphabet letters, and Thai characters) were significantly higher than that of the simple symbols. Hiragana consists of phonograms that do not have a specific meaning as a word, and it is difficult to associate Hiragana in a stimulus image with a particular word. Thus, subjects focused on the Hiragana, alphabet letters, and Thai characters and recognized them as hardwired salient objects rather than reading the characters as words. The degrees of visual saliency for the inserted characters and symbols calculated with the saliency map model are presented in Table 1. As Table 1 indicates, there were no significant differences in visual saliency between characters (Hiragana, alphabet letters, or Thai characters) and symbols. If the human attention is caused by the physical features represented by the saliency map model, the cumulative fixation ratios of the characters and symbols should be the same. However, the experimental results show that there were significant differences in the fixation ratio between characters and symbols. Thus, the characters will have a specific saliency that cannot be explained by the physical features used in the saliency map model.

**Table 1.** Means and standard deviations of visual saliency within the regions of inserted characters and symbols.

|  | Region A | Region B |
|---|---|---|
| Dataset 1 Mean (SD) | Hiragana 0.154 (0.088) | Symbols 0.152 (0.163) |
| Dataset 2 Mean (SD) | Alphabet 0.165 (0.096) | Symbols 0.152 (0.163) |
| Dataset 3 Mean (SD) | Alphabet 0.165 (0.096) | Hiragana 0.152 (0.085) |
| Dataset 4 Mean (SD) | Thai character 0.203 (0.094) | Symbols 0.192 (0.168) |
| Dataset 5 Mean (SD) | Thai character 0.203 (0.094) | Hiragana 0.152 (0.095) |

Next, we discuss the experimental results shown Figures 6 and 8. In this experiment, two types of characters were inserted into the different regions of the target image. In contrast to the comparison between characters and symbols, there were no significant differences in the cumulative fixation ratios between the Hiragana and alphabet letters or the Hiragana and Thai characters. Under this condition, the specific saliency of the respective characters contributed equally to evoking visual attention. These experimental results indicate that the degrees of visual saliency for the characters are almost the same irrespective of language.

## 5. Conclusions

In order to propose a highly accurate saliency map model, we investigated visual attention with respect to different types of characters. The experimental results showed that characters (Hiragana, alphabet letters, and Thai characters) have significant visual saliency. By contrast, no significant differences in visual saliency were found among Hiragana, alphabet letters, and Thai characters. As the experimental results indicated, the characters have significant visual saliencies independent of their familiarity.

In a future study, we plan to modify the saliency map model by considering the characteristics of visual saliency for character features.

**Author Contributions:** Conceptualization, H.T. and T.N.; methodology, H.T. and T.N.; software, T.N.; validation, H.T., T.N. and K.N.; formal analysis, H.T. and T.N.; investigation, T.N.; writing—original draft preparation, H.T.; writing—review and editing, H.T.; visualization, H.T. and T.N.; supervision, H.T.; project administration, K.N. All authors have read and agreed to the published version of the manuscript.

**Funding:** This research received no external funding.

**Institutional Review Board Statement:** The study was approved by the Ethics Committee of Toyama Prefectural University (H27-3; Approved: 9 June 2015).

**Informed Consent Statement:** Informed consent was obtained from all subjects involved in the study.

**Data Availability Statement:** The data presented in this study are available on request from the corresponding author. The data are not publicly available due to the ethical or privacy restrictions.

**Conflicts of Interest:** The authors declare no conflict of interest.

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
