# Peer review of "Characteristics of Visual Saliency Caused by Character Feature for Reconstruction of Saliency Map Model"

_2411-5150, 2021_

Round 1

Reviewer 1 Report

The current work investigates how the inclusion of text changes the deployment of attention to a scene and whether this can be accounted for by the low-level features of text or is influenced by higher-order processing, driven by the viewer’s familiarity with the language the text was written in. The authors tackle an interesting space that resides between what we refer to as strictly bottom-up saliency and more top-down influences on overt attention. Successful gaze prediction models often taken into consideration a bias towards stimuli such as text and faces. Accounting for this bias doesn’t seamlessly fit into bottom-up processes as often times the raw feature values of these stimuli are not particularly pop-out in the way we think of other salient regions of a scene (e.g. a red ball in a green field). However they also don’t fit into more restricted definitions of top-down drivers of attention because they are not strictly part of any current task set (at least not one imposed on by the researcher). For this reason I think this area of research that the authors are pursuing is a valuable one. However, there are a lot of critical details from this experiment that are missing, without which I can’t recommend this paper be published. Below are some of the points that would need to be addressed (I’ve tried to break them out into the various parts of the paper to be helpful): Introduction 1) I think the introduction would generally be improved by defining some of the terms used. I think people mean slightly different things when they use the same terms and defining things from the onset can help clear up that confusion. For instance, the focus of the paper is on saliency maps which the author says are constructed from feature maps (orientation, color, etc). I think this is specific to the Itti & Koch model which the authors cite. There are however a plethora of deep neural net models that are pitched as saliency models and yet do not have these predefined maps (though some layers of the model may correspond with these feature maps). Graph-Based Visual Saliency model is one that comes to mind. There are other models such as Meaning maps (see Henderson’s work) that take into account the “meaning” of given region. This moves slightly away from pure bottom-up saliency but is relevant to the author’s work as both are using free gaze as their test metric. I think it’s reasonable to limit your definition of saliency to how regions of the visual field pop-out on various feature dimensions (e.g. Itti model) as other saliency models train purely on eye tracking data which, this work shows is not purely driven by feature differences. However there needs to be clear delineation made and some additional background work acknowledged. 2) Similar to the previous point it would be good if the authors provided a working definition for top-down and bottom-up attention. As their paper addresses, there is a grey area and so it would be good if they set their own definition up front. 3) In the section related work there are no citations of any related work for visual attention despite that being an important component of the current work. One paper that may be of interest to the authors is Rufin VanRullen’s work on hard wired stimuli which talks about how training on given stimulus can change the way it’s searched for. Obviously the participants are not explicitly searching for text here but as reading is something that we devote a profound amount of time to train on, this paper may benefit from some discussion on that changes a stimulus’ processing. Methods 1) I am very confused about the text stimuli used in this experiment. Partly I think this is due to inconsistent naming throughout the manuscript and so may be corrected with just some editing. At various points the authors refer to using the following text stimuli: HIRAGANAs, alphabets, Thai characters, simple symbols. Are these discrete categories? I’m afraid part of this is ignorance on my part of how characters differ from symbols and alphabets. I think (based on a sample of 1, myself) that a western audience would need a break-down of what each of these stimulus categories are and how they relate to one another, if at all. The reader would also benefit from a figure that showed examples from each text stimulus type. 2) Was any analysis done to compare the features of one text category from another. My takeaway was that the purpose of the work is to see whether it is familiarity with a language over the low-level feature properties of its text that determines if the eyes go there. If that’s the case then there has to be some discussion as to the low-level similarities of these texts 3) What was the language background of the participants? The authors mention that they did not speak Thai but would they be know or recognize it’s script? If the authors are making claims about the effect of familiarity, we need some quantified familiarity of the participants when they started this experiment. Also, is the alphabet or simple symbols related at all to a language? 4) It was unclear whether the same images were used in each of the “datasets” or if different images were used. This obviously has consequences for interpreting the results. 5) The authors say in section 3.1.3 that they limited the images by their valence and arousal score in order to “suppress top down attention”. I’m not sure what this means (see point 2 under introduction about providing definitions). 6) In section 3.2 the authors make some mention of the difference between where cover attention goes and where the eyes move to. I’m not sure I agree with all of the claims made there but a simpler approach would be to just say that you are measuring “overt attention” (i.e. where the eyes move as a method of selection). Results 1) The authors report a fixation ratio where the denominator is the number of times the eyes are presented the target and the numerator is the number of times the eyes fixate on the target. Does this count repeated fixations on the target? 2) There’s a mentioning that the order was different for fixating on various text stimuli but there is no statistical test to verify this 3) All subsequent stats reporting should have the full report (t-values etc not just p-values) and effect sizes should be included.

Author Response

Response to Reviewer 1 Comments

The current work investigates how the inclusion of text changes the deployment of attention to a scene and whether this can be accounted for by the low-level features of text or is influenced by higher-order processing, driven by the viewer’s familiarity with the language the text was written in. The authors tackle an interesting space that resides between what we refer to as strictly bottom-up saliency and more top-down influences on overt attention. Successful gaze prediction models often taken into consideration a bias towards stimuli such as text and faces. Accounting for this bias doesn’t seamlessly fit into bottom-up processes as often times the raw feature values of these stimuli are not particularly pop-out in the way we think of other salient regions of a scene (e.g. a red ball in a green field). However, they also don’t fit into more restricted definitions of top-down drivers of attention because they are not strictly part of any current task set (at least not one imposed on by the researcher). For this reason I think this area of research that the authors are pursuing is a valuable one. However, there are a lot of critical details from this experiment that are missing, without which I can’t recommend this paper be published. Below are some of the points that would need to be addressed (I’ve tried to break them out into the various parts of the paper to be helpful):

We wish to express our appreciation to the reviewer for your insightful comments on our paper. These comments have helped us significantly improve our paper. We show the reply to reviewer’s comments as follows.

Introduction

  • I think the introduction would generally be improved by defining some of the terms used. I think people mean slightly different things when they use the same terms and defining things from the onset can help clear up that confusion. For instance, the focus of the paper is on saliency maps which the author says are constructed from feature maps (orientation, color, etc). I think this is specific to the Itti & Koch model which the authors cite. There are however a plethora of deep neural net models that are pitched as saliency models and yet do not have these predefined maps (though some layers of the model may correspond with these feature maps). Graph-Based Visual Saliency model is one that comes to mind. There are other models such as Meaning maps (see Henderson’s work) that take into account the “meaning” of given region. This moves slightly away from pure bottom-up saliency but is relevant to the author’s work as both are using free gaze as their test metric. I think it’s reasonable to limit your definition of saliency to how regions of the visual field pop-out on various feature dimensions (e.g. Itti model) as other saliency models train purely on eye tracking data which, this work shows is not purely driven by feature differences. However there needs to be clear delineation made and some additional background work acknowledged.

Response : Thank you for your valuable comment. As you indicated, the definition of “saliency map” is ambiguous. The saliency map focusing on our paper is Itti’s model which is constructed with the low-level features, i.e. intensity, color, orientation. However, we address the saliency of character (text) which cannot be expressed by Itti’s model. We introduce other saliency map model other than Itti’s model and add the detailed definition of saliency map. The sentences added in the manuscript are shown as follows.

(Page 1, Sec.1 2nd Paragraph, Line 6)

A plethora of saliency map models that are based on not only bottom-up attention but also on top-down attention or the fusion of both types have been proposed. A graph-based saliency map (GBVS) is constructed by introducing a fully connected graph into Itti’s model [2]. The proto-object saliency map model employs a biologically plausible feature [3][4]. The hybrid model of visual saliency was developed using low, middle, and high-level image features [5]. In recent years, deep learning based models have been proposed. These models express the characteristics of bottom-up attention [6][7], hybrid bottom-up and top-down attention [8], eye fixation [9], and sequential eye movements [10][11]. However, instead of an image feature, a meaning map that utilizes the meaning information in a scene has been proposed [12]. Comparing the prediction accuracy of human attention obtained from the meaning map and GBVS, the spatial distribution of attention is similar in both the methods. However, by controlling the relationship between meaning and saliency, the meaning map sufficiently expresses the attention guidance compared with GBVS. In this study, the model derived using the low-level features is deal with as a saliency map.

  • Similar to the previous point it would be good if the authors provided a working definition for top-down and bottom-up attention. As their paper addresses, there is a grey area and so it would be good if they set their own definition up front.

Response : Thank you for your suggestion. In this study, we regard the bottom-up attention induced by the visual saliency which is constructed by the low-level features, such as the intensity, color, orientation. On the other hand, the top-down attention is task-driven and dependent on the prior knowledge of each individual. Furthermore, we consider that the attention to the character exists between bottom-up and top-down attention. Thus, the character cannot always be considered as the objects which are represented using bottom-up saliency. We added the explanation about the definition of attention and the attention paid to the character.

(Page 1, Sec.1 1st Paragraph, Line 6)

This study focuses on the visual attention derived from the gaze information. The visual attention is divided into two categories: top-down and bottom-up attention. Bottom-up attention is defined by the attention to a visual salient region which is represented by low-level features, such as the intensity, orientation, and color. Top-down attention is task-driven and dependent on the prior knowledge of each individual.

(Page 2, Sec.1 4th Paragraph, Line 3)

Because characters contain high-level semantic information, they cannot always be considered as the objects which are represented using bottom-up saliency. In this study, we considered that the attention paid to the characters exists between the bottom-up and top-down attention.

  • In the section related work there are no citations of any related work for visual attention despite that being an important component of the current work. One paper that may be of interest to the authors is Rufin VanRullen’s work on hard wired stimuli which talks about how training on given stimulus can change the way it’s searched for. Obviously the participants are not explicitly searching for text here but as reading is something that we devote a profound amount of time to train on, this paper may benefit from some discussion on that changes a stimulus’ processing.

Response : Thank you for your invaluable suggestion. As you explained, Rufin VanRullen describes that familiar objects can be recognized without attention. The characters are assumed to be familiar objects because we gain knowledge of the character through training over time. Thus, attention to characters is not necessarily described as a saliency map obtained from the low-level features. We added the explanation about the hardwired binding process in Section 2.1.

(Page 2, Sec.2.1, Line 5)

The recognition of natural and familiar objects and the search for frequent stimulations have been conducted using a hardwired binding process that occurs without attention [17]. In the task of face recognition or gender identification from faces, the importance of attention is low for task implementation because the face is considered a familiar object [18][19]. Because we gain knowledge of the characters through training over time, the characters are assumed to be familiar objects. Thus, we consider that character recognition is performed through a hardwired process, which does not always involve attention. In other words, attention to characters is not necessarily described as a saliency map obtained from the low-level features.

Methods

  • I am very confused about the text stimuli used in this experiment. Partly I think this is due to inconsistent naming throughout the manuscript and so may be corrected with just some editing. At various points the authors refer to using the following text stimuli: HIRAGANAs, alphabets, Thai characters, simple symbols. Are these discrete categories? I’m afraid part of this is ignorance on my part of how characters differ from symbols and alphabets. I think (based on a sample of 1, myself) that a western audience would need a break-down of what each of these stimulus categories are and how they relate to one another, if at all. The reader would also benefit from a figure that showed examples from each text stimulus type.

Response : As you indicated, the audience cannot judge whether the category of each character is quite different. An example of characters inserted into stimulus images is shown as the figure 1 of the revised manuscript. We added the explanation about the relationship among Hiragana, alphabet letters, Thai characters, and simple symbols.

(Page 3, Sec.3, Line 4)

An example of the characters and symbols inserted into visual stimuli is shown in Fig.1. Figures 1(a) – (d) show Hiragana, alphabet letters, Thai characters, and simple symbols. As shown in this figure, these characters and symbols are classified into the different categories because their shapes are quite different from each other.

  • Was any analysis done to compare the features of one text category from another. My takeaway was that the purpose of the work is to see whether it is familiarity with a language over the low-level feature properties of its text that determines if the eyes go there. If that’s the case then there has to be some discussion as to the low-level similarities of these texts.

Response : As you indicated, the purpose of this study is to investigate whether the familiarity of a language influences the characteristics of fixation. We consider that the low-level feature property of each character is almost the same because the size of the inserted character is sufficiently small compared with the size of the stimulus image. We added the explanation about the low-level feature property of characters inserted in the stimulus image.

(Page 4, Sec.3.1.3 2nd Paragraph, Line 6)

The low-level feature properties of the characters and symbols may give rise to differences in the characteristics of attention. Therefore, the sizes of the characters and symbols inserted into the stimulus image are sufficiently small compared with the size of the stimulus image itself.

  • What was the language background of the participants? The authors mention that they did not speak Thai but would they be know or recognize it’s script? If the authors are making claims about the effect of familiarity, we need some quantified familiarity of the participants when they started this experiment. Also, is the alphabet or simple symbols related at all to a language?

Response : From the preliminary questionnaire, all participants could not read and write in Thai. By contrast, they could read and write in English and recognized “simple symbol” as the symbol. The familiarity of each character and symbol is not different in each individual. We added the explanation for the language background of participants and the effect of familiarity.

(Page 4, Sec.3.1.2 1st Paragraph, Line 5)

The native language of the subjects was Japanese. All the subjects could read and write in English. Although the subjects could not recognize Thai characters as being Thai, they recognized them as characters of a particular language rather than symbols. The individual difference in familiarity between the characters and symbols could not be confirmed from the results of the questionnaire, however.

  • It was unclear whether the same images were used in each of the “datasets” or if different images were used. This obviously has consequences for interpreting the results.

Response : The images used in each dataset are the same. In the revised manuscript, we described that the same images are used in all datasets.

(Page 4, Sec.3.1.2 2nd Paragraph, Line 8)

Among Datasets 1-5, the images used as visual stimuli are of the same combination.

  • The authors say in section 3.1.3 that they limited the images by their valence and arousal score in order to “suppress top down attention”. I’m not sure what this means (see point 2 under introduction about providing definitions).

Response : The fixation property of each participant would be changed by emotions that are induced by stimulus images. Thus, we limited the valence and arousal scores to keep the participant’s emotion as the neutral condition. The representation of experiment condition about stimulus image was modified as follows.

(Page 4, Sec.3.1.3 1st Paragraph Line 8)

To suppress the individual variation in the top-down attention feature caused by the emotion induced, we adopted these image selection criteria.

  • In section 3.2 the authors make some mention of the difference between where cover attention goes and where the eyes move to. I’m not sure I agree with all of the claims made there but a simpler approach would be to just say that you are measuring “overt attention” (i.e. where the eyes move as a method of selection).

Response : Thank you for your suggestion. We conducted the experiment as the free-viewing task. As you suggested, we explained that the gaze caused by overt attention was measured in the experiment.

(Page 5, Sec.3.2 1st Paragraph Line 5)

we simply measured the eye movement that occurred during an overt attention.

Results

  • The authors report a fixation ratio where the denominator is the number of times the eyes are presented the target and the numerator is the number of times the eyes fixate on the target. Does this count repeated fixations on the target?

Response : In this analysis, the repeated fixation was counted as “one time” even if the fixation was occurred on the target many times. The following explanation was added in the revised manuscript.

(Page 6, Sec.3.3 1st Paragraph Line10)

Note that the repeated fixations on the characters or symbols are not counted.

  • There’s a mentioning that the order was different for fixating on various text stimuli but there is no statistical test to verify this.

Response : As you indicated, We deleted the description about the difference of fixation related to the order.

  • All subsequent stats reporting should have the full report (t-values etc not just p-values) and effect sizes should be included.

Response : We added the p-value, t-value, and effect size “d” in the revised manuscript.

Reviewer 2 Report

  1. Kindly carry out an extensive English language review of the manuscript
  2. Kindly rewrite the introduction and literature review. A richer introduction to the topic need to be presented, as well as a larger number of literature sources that are more recent need to be included

Author Response

Response to Reviewer 2 Comments

We wish to express our appreciation to the reviewer for your insightful comments on our paper. These comments have helped us significantly improve our paper. We show the reply to reviewer’s comments as follows.

  • Kindly carry out an extensive English language review of the manuscript

Response : English of our manuscript was checked by the English editing service.

  • Kindly rewrite the introduction and literature review. A richer introduction to the topic need to be presented, as well as a larger number of literature sources that are more recent need to be included

Response : As you suggested, we have rewritten the introduction. The contents of the Introduction were expanded and literatures were added. The additional contents are as follows.

  • Introduction of saliency map model other than Itti’s model
  • Definition of bottom-up and top-down attention
  • Characteristics of attention paid to the characters
